# Park Visitors and Birds Connected by Trade-Offs and Synergies of Ecosystem Services

**DOI:** 10.3390/ani15172619

**Published:** 2025-09-06

**Authors:** Yichao Chen, Liyan Zhang, Zhengkai Zhang, Siwei Chen, Bei Yu, Yu Wang

**Affiliations:** College of Landscape Architecture and Art, Northwest A&F University, Yangling 712100, China; ycchen@nwafu.edu.cn (Y.C.); zhangly_2000@163.com (L.Z.); siweic@nwafu.edu.cn (S.C.); yb.1206@nwafu.edu.cn (B.Y.); wang_yu_@nwafu.edu.cn (Y.W.)

**Keywords:** ecosystem services, trade-offs and synergy, bird, urban park

## Abstract

Parks provide essential habitats for birds and a relaxing space for residents, which could cause a spatial overlap with natural competition between humans and birds. Previous studies found that different landscape patterns, vegetation structures, and management strategies for green spaces lead to trade-offs among a variety of ecosystem services. However, it remains unclear how parks could coordinate visitor–bird relationships from a perspective on ecosystem services through landscape design and management. To fill this gap, we conducted a bird census and social surveys from July to November 2023, at Xingqing Palace Park in Xi’an, China. We analyzed the spatial relationships among the park’s cultural ecosystem services (CES), the park’s supporting ecosystem services (SES), and bird plumage color CES. We found that high-coverage vegetation areas along main roads promoted a synergistic effect, which benefited visitors’ appreciation of the park’s CES, bird roosting activity, and the avian CES. Meanwhile, there were trade-offs between the aesthetic value of the park’s CES and SES and bird plumage color CES, which is primarily present in the plaza with noise levels > 70 dB. Our findings indicated that planting large trees along park boundaries to expand the ecological buffer zone and adding small waterbodies will help to improve both avian CES and park CES.

## 1. Introduction

Wildlife and humans coexist in an increasingly urbanized world, where over half of the global population resides in urban areas [1]. Green spaces are essential infrastructure for the sustainability of urban ecosystems, delivering a variety of ecosystem services (ES), including cultural ecosystem services (CES), regulating ecosystem services (RES), and supporting ecosystem services (SES). Regulating services are the benefits obtained from the regulation of ecosystem processes; cultural services are the nonmaterial benefits people obtain from ecosystems; supporting services are those that are necessary for the production of all other ecosystem services [2]. Numerous studies have shown that the CES provided by urban parks contribute significantly to human health and well-being. These spaces offer opportunities for locals and visitors to connect with nature, engage in recreation, and enjoy leisure activities [3,4]. Additionally, urban parks are crucial habitats for birds, as they provide a variety of habitats for breeding, foraging, and roosting, which is reflected in species and functional diversity. Birds also offer a range of ecosystem services, including provisioning, regulating, supporting, and cultural services [5]. The most immediately perceived of these is the cultural service, often associated with bird traits, such as plumage color and song, that provide aesthetic and recreational benefits to people. Among these traits, plumage color is one of the most highly appreciated characteristics [6,7]. However, spatial overlap between humans and birds inevitably leads to spatial overlap with natural competition [8]. Visitors may increasingly encroach upon sensitive areas, such as woodlands, natural pools, and lakes, which may severely disturb the habitats of birds in urban parks [9]. Moreover, as co-users of park resources, birds and humans often have distinct and sometimes conflicting demands, especially for vegetation and water [10]. Consequently, trade-offs may arise between the CES provided by urban parks and the SES utilized by birds, as well as between the CES provided by the parks and those contributed by birds themselves. Thus, understanding the relationships between visitors and birds from an ecosystem services perspective can help mitigate competition arising from spatial overlap, promote bird conservation, and enrich the visitor experience.

The trade-offs and synergies of ecosystem services are closely linked to urban sustainability and human well-being, attracting significant attention [11,12]. Some studies have explored the link between ES, biodiversity protection, and human well-being at the city-regional scale. The study found that mental health and bird diversity were positively influenced by vegetation structure and habitat heterogeneity, while bird species diversity did not have a direct effect on mental health [10]. Similarly, Paudel and States revealed that although lawns were more highly valued by visitors, meadows were more effective in providing regulating and supporting services [13]. Other research has investigated the spatial trade-offs and synergies of ES within parks, revealing that the spatial patterns of CES were linked to different landscape features [14]. The CES of green spaces can be categorized into two types based on their providers: services directly provided by parks and services indirectly provided by biodiversity in green spaces [2]. Therefore, the spatial trade-offs and synergies of SES and CES delivered by biodiversity remain unclear.

The trait-based ES framework, originally developed in functional ecology, aims to elucidate the relationship between socio-ecological processes and the ES provided by biodiversity. It also serves as an effective tool for studying CES associated with biodiversity in green spaces [15,16]. In this framework, some functional traits related to environmental tolerance, habitat needs, and stress responses are categorized as response traits. Other traits associated with the provision of ecosystem services are classified as effect traits. Based on this framework, several studies have identified relationships between birds’ responses to the environment and vegetation cover [17], vegetation structure [18], and water bodies [19] in their habitats. Other studies have focused on the CES provided by bird effect traits, revealing that colorful species can offer pleasurable and aesthetic experiences [20] and bird vocalizations can reduce psychological stress while enhancing nature-based experiences and satisfaction with green spaces [21]. The majority of these studies have concentrated on large-scale settings, such as national parks [14] and urban regions [16,22]. In contrast, few studies have been conducted at the neighborhood level or within parks, which hinders efforts toward human-scale sustainability. Drawing on these insights, we developed a trait-based ecosystem service framework tailored to the scope of this study (Figure 1).

In this study, we selected Xingqing Palace Park in Xi’an, the city’s largest comprehensive urban park, as the study area. The park boasts a long history, rich landscape composition, and a diverse range of visitors. We quantified and mapped park SES and avian CES using the MaxEnt model, then combined SolVES modeling with participatory GIS (PPGIS) to measure and trace park CES. Finally, we analyzed the trade-offs and synergies between various ES using bivariate spatial autocorrelation. The research objectives are as follows: (1) What are the trade-offs and synergies between SES and CES provided by urban green spaces, as well as between the CES provided by birds inhabiting the park? (2) How do the spatial characteristics and human activities of the urban park influence these relationships? The outcomes of this research will support informed landscape management and spatial planning in urban parks, which will promote biodiversity conservation and recreational opportunities, ultimately contributing to the sustainability of urban ecosystems.

## 2. Materials and Methods

### 2.1. Study Area

The study area is located in the Beilin District of Xi’an City (34°15′13.32″ N, 108°58′7.32″ E), which experiences a warm-temperate continental monsoon climate with four distinct seasons, including cold winters and hot summers. The average annual temperature ranges from 11.1 to 13.3 °C. The park, which was established in 1958 on the site of the Tang Dynasty’s Xingqing Palace, covers an area of 492,000 m^2^ (100,000 m^2^ of which is water) (Xi’an Xingqing Palace Park official website: https://www.xingqinggong.com/ (accessed on 17 July 2025)). It is surrounded by residential neighborhoods, government offices, and educational institutions, including kindergartens, primary and secondary schools, as well as three university campuses. With a diverse array of internal landscapes and a dynamic cultural environment, the park holds significant historical and cultural value for Xi’an City and serves as a popular destination for leisure and recreation for local residents (Figure 2).

### 2.2. Data Sources

#### 2.2.1. Bird Census

The bird survey was conducted using six line transects, with a total length of 2196 m and an average length of 366 m per transect, strategically designed to cover the park’s entrances, core areas, edge zones, various habitat types, and main visitor activity areas. Surveyors walked at a speed of 2.0 km/h, recording bird species, counts, and GPS locations observed or heard within a 25-m range on either side of the transect (including overflying individuals). Each transect had a minimum spacing of 150 m between transects to avoid duplicate counts. The survey was carried out from July to November 2023, with twice-monthly surveys in September and November (migration season) and monthly surveys in July and August (breeding season). Equipment included 8 × 42 mm binoculars, electronic monoculars, and digital cameras. Surveys were conducted during peak bird activity hours (two hours after sunrise) under clear, windless conditions. To ensure data consistency, each survey involved two teams conducting synchronous surveys by respectively selecting one transect line. All transect surveys were completed within the same day, with the survey sequence being randomized prior to each investigation. For bird count, we adopted the maximum count retention method, whereby each species was surveyed twice per month, and the higher count from the two surveys was retained as the monthly record.

#### 2.2.2. Cultural Activity Survey

The research data were collected through a field survey. We ensured that no personal privacy or mental health issues were involved in the study, and all respondents signed an informed consent form. The questionnaire consisted of four sections: the first section assessed visitors’ leisure experiences (Appendix A); the second section focused on understanding visitors’ perceptions and attitudes toward birds; the third section gathered information on visitors’ recreational behaviors and demographics, such as age, gender, and educational background; and the fourth section asked visitors to assign social values to CES and annotate their value assignments. Based on the pre-survey, this section identified seven distinct social values for CES, as shown in Table 1, and respondents were asked to “spend” or “allocate” a hypothetical RMB ¥100 among these seven CES. The survey was conducted between 15 July 2023 and 12 November 2023 in clear, sunny weather, ideal for outdoor activities. A validity rate of 98.79% was achieved, with 163 valid surveys collected out of a total of 165 responses. Among these, responses that contained no usable information or incomplete data after collection were categorized as invalid questionnaires.

#### 2.2.3. Environmental Survey

As shown in Figure 3, the SolVES and MaxEnt models require the integration of various environmental data, in addition to the analysis of social value points and regional boundaries, to fully describe the environmental characteristics of the study area. To describe the environmental characteristics of the study area for SolVES and MaxEnt modeling, while working within equipment constraints, this study has made every effort to select the maximum number of feasible environmental factors for comprehensive analysis. We collected data on land use and land cover (LULC), visible band difference vegetation index (VDVI), distance to road (DTR), distance to water (DTW), digital surface model (DSM), and noise level (NL) as the environmental variables for the study area. The spatial resolution of the data for each of these environmental indices was five meters. The data sources and processing procedures are summarized in Table 2. The boundary of the study area was captured via aerial photography using a DJI Air 2S drone DJI Air 2S drone (SZ DJI Innovation Technology Co., Ltd., Shenzhen, Guangdong, China) (flight altitude: 200 m, camera angle: 90°), and the images were subsequently stitched using Pix4D 4.5.6 software to generate a planar map of the park. For the noise survey, three rounds of measurements were conducted. Along each avian survey transect, 7–8 sampling points were evenly distributed at 50-m intervals. Noise levels were measured at each point using a sound level meter during the same time period. Measurements were taken by holding the instrument stationary at 1.5 m above the ground, with the microphone oriented vertically downward (covered by a windscreen). The maximum noise level (NL) and GPS coordinates were recorded for each sampling point.

### 2.3. Analysis

#### 2.3.1. Assessment of Ecosystem Services

This study focused on three aspects of ES: park cultural ecosystem services (park CES), which reflect the park’s ability to provide aesthetic, recreational, and other values; park supporting ecosystem services (park SES), which pertain to the park’s capacity to sustain bird diversity and provide habitats for birds; and avian cultural ecosystem services (avian CES), which represent the aesthetic benefits offered to visitors by birds inhabiting the park. We used the SolVES model to assess park CES and the MaxEnt model to assess park SES and avian CES.

SolVES is a GIS application developed by the U.S. Geological Survey (USGS) to assess the perceived social values, or CES, such as aesthetics and recreation, for different stakeholder groups [24]. Our study utilized the ecosystem services social-values module and the value mapping module of the SolVES model (Figure 4). Table 2 presents the variables used in the SolVES model. The model generates value index maps for different park CES, with the index ranging from 1 to 10, where a higher index value indicates greater value as perceived by visitors.

MaxEnt is an ecological model based on ecological niche theory, which requires species distribution and spatial raster data to project the ecological needs of species and estimate their potential distribution patterns across different times and spaces [25]. It has been shown to be applicable to small-scale studies [26]. This study primarily focuses on the spatial association between visitors and birds within the park. As feeding guilds reflect how birds utilize park resources, we selected this trait to represent the supporting services provided by the park. We used *A Handbook of the Birds of China* to categorize the modeled bird species into four feeding guilds: omnivorous, insectivorous, granivorous, and piscivorous [27]. We created binary maps for each feeding guild, with suitability index values >0.6 coded as a 1 (“suitable”), then combined binary maps using the raster calculator in ArcGIS to obtain a single map representing the estimated richness of bird feeding guilds across our study area. We considered this map as representing avian habitat requirements across guilds and, thus, the distribution pattern of park SES. (Figure 5). Furthermore, we used bird plumage color CES as one of the criteria to assess avian CES, as it is directly perceptible to most visitors. Based on data from the second part of the questionnaire, titled “*The following questions about your attitudes toward birds in the park*”, 61.7% of respondents reported being attracted to birds primarily by their plumage color, a proportion significantly higher than that for bird size, body size, or behavior. Therefore, this study selected plumage color characteristics as the representative indicator of CES provided by birds. We extracted the plumage color traits of the birds involved in the model, as referenced in A Handbook of the Birds of China [27], and quantified them using the formula proposed by Swartz et al., employing the raster calculator in ArcGIS to determine how habitat-effect trait associations influence bird plumage color CES. The following are the specific formulas:(1)[CSETwarm+CSETcool]−[CSETdull+CSETblack]

Warm (red, orange, yellow) and cool (blue, purple, green) plumage hues are generally more aesthetically appealing than black and dull (gray, brown, tan) plumage hues [7,28]. The spatial distributions of CES provided by bird effect traits were subsequently characterized.

#### 2.3.2. Bivariate Spatial Autocorrelation

We set 5 m × 5 m grid cells to construct the spatial weight matrix. Additionally, we employed both global and local bivariate spatial autocorrelation in GeoDa to investigate the spatial correlation and aggregation between park SES and park CES, as well as between bird plumage color CES and park CES. These were computed using the following formula:(2)$$\mathrm{Global}\;\mathrm{Moran}’s\;I=\frac{n\sum_{i=1}^{n}\sum_{j=1}^{n}W_{ij}\left(x_{i}-\bar{x}\right)\left(x_{j}-\bar{x}\right)}{n\sum_{i=1}^{n}\sum_{j=1}^{n}W_{ij}{\sum_{i=1}^{n}\left(\bar{x_{i}}-x\right)}^{2}}$$
where *I* is the global autocorrelation index, *n* is the number of geographic samples, *x_i_* and *x_j_* are the observed values of units *i* and *j*, and *W_ij_* is the spatial weight matrix. The range of global Moran’s *I* values is [−1, 1], with a value of zero indicating a generally random geographic distribution pattern. A Moran’s *I* value greater than zero indicates positive spatial correlation, while a value less than zero indicates negative spatial correlation. Local Moran’s *I* introduces a local spatial autocorrelation index based on the Z-test (Local Indicators of Spatial Association, LISA), and LISA plots are generated. In this case, trade-offs are represented by both “high-low” and “low-high” aggregations, while “low-low” and “high-high” aggregations indicate synergistic attenuation and synergistic promotion, respectively.

#### 2.3.3. Geographical Detector

Geographic detectors are employed to examine the interaction effects of explanatory variables and potential influencing factors [29]. We used this model to assess the strength of each factor in Table 2 in influencing the trade-offs and synergies between park SES and park CES, as well as between bird plumage color CES and park CES. A stronger correlation between the explanatory variable (*X*) and the explained variable (*Y*) occurs when their geographic distributions are similar. The degree to which *X* explains *Y* is expressed as the deterministic power factor *q*, which quantifies the similarity between the spatial distributions of *X* and *Y* [28]. The following formulas were used to determine the *q* values:(3)$$q=1-\frac{1}{N\sigma^{2}}\sum_{h=1}^{n}N_{h}\sigma_{h}^{2}$$
where *q* is the detection power of the detection factor, *h* = 1,…, *n* represents the strata of the variable or factor; *N_h_* and *N* are the number of cells in stratum *h* and the total area, respectively; and *σ_h_^2^* and *σ^2^* are the variances of the Y-values in stratum *h* and the entire area, respectively. The value of *q* ∈ [0, 1], with higher *q* values indicating stronger associations.

## 3. Results

### 3.1. Bird Census and Ecosystem Services Related to Avian Traits

#### 3.1.1. Bird Community Composition

A total of 745 individuals, representing 25 species, were recorded. Passerine birds dominated the community, with 17 species and 649 individuals. The five most common species were the Azure-winged Magpie (*Cyanopica cyanus*), Tree Sparrow (*Passer montanus*), White-browed Laughingthrush (*Garrulax sannio*), Light-vented Bulbul (*Pycnonotus sinensis*), and Bare Swallow (*Hirundo rustica*). A complete list of all species can be found in Appendix A.

#### 3.1.2. The Park Supporting Ecosystem Services Related to Bird Feeding Guilds

The MaxEnt model demonstrated successful AUC accuracy testing for all species (mean AUC = 0.767, Appendix A). The spatial distribution of park SES was concentrated along the main roads and woodlands. In particular, the woodlands to the north and east supported a greater number of bird-feeding guilds, providing better SES. In contrast, areas near water bodies and along the western and southern boundaries of the park supported fewer bird-feeding guilds, suggesting fewer SES in these locations (Figure 6a).

#### 3.1.3. The Avian Cultural Ecosystem Services from Bird Effect Traits

In general, the spatial pattern of bird plumage color CES produced more positive than negative effects. In the eastern part of the park and along the main roads, bird plumage colors had more beneficial impacts. However, in the northwestern woodlands of the park, they had some negative effects (Figure 6b).

### 3.2. Cultural Ecosystem Services of the Park

#### 3.2.1. Sample Description

A total of 163 valid questionnaires was collected. A total of 67.5% of total respondents visited Xingqing Palace Park at least once a month, and 87.1% visited the park for leisure and exercise, in line with the park’s orientation. Meanwhile, the responders who were satisfied with and recognized each indicator with Xingqing Palace Park accounted for 74.2% of the total survey respondents. For the rest of the information see Appendix A.

#### 3.2.2. Evaluation Analysis and Spatial Clusters of Park Cultural Ecosystem Services

Due to the fact that maximum value index was vastly different among the seven CES (Table 3), based on the clustering pattern (with an average nearest neighbor ratio less than 1 and a significantly large absolute Z-value) and a maximum value index ≥5, recreational value, aesthetic value, social relationship value, and historical heritage value were selected for mapping and subsequent correlation analysis. This approach allows for an objective investigation of the relationship between park CES, park SES, and bird plumage color CES under the environmental factors in the park.

The spatial cluster results show that most of the hard-paved plazas in the eastern part of the park were rich in all types of CES. In contrast, other areas had relatively lower levels of CES (Figure 6c). Specifically, areas with a recreational value above 6 were evenly distributed throughout the park, while areas with values greater than 8 were concentrated on the eastern side (Figure 6d). The spatial pattern of social relationship value closely resembled that of recreational value. However, areas with the highest value index were generally lower than those for recreational value (Figure 6e). Areas with an aesthetic value greater than 6 were primarily located in the water bodies surrounding the island, as well as on the island itself (Figure 6f). The architectural landscape had high historical value, with the highest value index of 6 (Figure 6g).

### 3.3. Trade-Offs/Synergies and Factors Among ES

#### 3.3.1. The Relationships and Factors Between Park Supporting Ecosystem Service and Park Cultural Ecosystem Services

Bivariate spatial autocorrelation revealed a negative correlation between park SES and aesthetic value (Moran’s *I* = −0.164), while park SES positively correlated with recreational and social relationship values (Moran’s *I* = 0.128, Moran’s *I* = 0.101). Other relationships were insignificant. A higher absolute value of Moran’s *I* indicates a stronger correlation between the relationship pairs. Therefore, relationships with an absolute Moran’s *I* value greater than 0.1 were further analyzed in subsequent local spatial autocorrelation. The LISA diagrams show that the relationships between park SES and park CES, as well as bird plumage color CES and park CES, exhibit significant spatial aggregation (Figure 7). The trade-offs predominantly affect the spatial relationship between park SES and aesthetic value (Figure 7a). Trade-off areas were mainly the residents’ fitness plazas, which were significantly impacted by human activity, and the waterbody center. Meanwhile, synergistic promotion areas were distributed along the roadside, while synergistic attenuation areas were located at the western and southern boundaries of the park. Notably, the spatial relationship between trade-offs/synergies of park SES and recreational value, as well as social relationship value, followed similar patterns. The proportion of spatial synergies between these two pairs of ecosystem services was higher than that of spatial trade-offs (Figure 7b,c). Synergistic promotion areas were primarily along the roadside and near park entrances, while synergistic attenuation areas were found at the western and southern park boundaries, as well as within the waterbody. In contrast, trade-offs were identified at the center of the waterbody and in the architectural areas. Interestingly, the synergies between park SES and recreational value covered a larger area than those between park SES and social relationship value, while the trade-offs were relatively smaller.

The results from the geodetector analysis indicate that NL (*q* = 0.30) has a strong impact on the relationships between park SES and aesthetic value (Figure 8). The trade-offs/synergies between park SES and recreational value, as well as park SES and social relationship value, are weakly influenced by individual factors.

The interactions between the factors influencing trade-off/synergistic relations among ecosystem services are shown in Figure 7. All interactions involve two-factor enhancement, suggesting that two-factor interactions have greater explanatory power than a single factor. For the trade-offs/synergies between park SES and aesthetic value, the interaction between NL and LULC, as well as NL and DTR, is the strongest, with an impact value of 0.48. This is followed by NL and DTW, and NL and DSM, both with impact values exceeding 0.35 (Figure 9a). Regarding the relationships between park SES and recreational value, as well as park SES and social relationship value, the strongest interactions were between NL and LULC, with impacts above 0.30 (Figure 9b,c).

#### 3.3.2. The Relationships and Factors Between Bird Plumage Color Cultural Ecosystem Service and Park Cultural Ecosystem Services

Bivariate spatial autocorrelation indicates that bird plumage color CES related to recreational value had a positive correlation (Moran’s *I* = −0.123), while bird plumage color CES related to aesthetic value exhibited a negative correlation (Moran’s *I* = −0.173). The remaining aspects showed little correlation. According to Lisa’s diagrams, the spatial pattern of trade-offs/synergies for bird plumage color CES related to recreational value reveals that the proportion of spatial synergies was greater than the spatial trade-offs (Figure 7d). The synergistic promotion areas appeared along main roads with vegetation cover. The synergistic attenuation areas were similar to other ES elements, appearing at the park’s boundary and within the waterbody. The trade-offs were found in the impervious surface areas and the waterbody center. In addition, the spatial pattern and space distribution reveals that the spatial trade-offs were to be a priority (Figure 7e). Unlike the trade-off between bird plumage color CES and aesthetic value coverage on a large scale, synergistic promotion is reflected in the small range of spatially fragmented areas.

Among the six driving mechanisms analyzed, NL had the most significant impact on the trade-offs/synergies between bird plumage color CES and recreational values. NL also influenced the relationship between bird plumage color CES and aesthetic value. Furthermore, DTR exhibited a notable effect on the trade-offs/synergies between bird plumage color CES and aesthetic value, as illustrated in Figure 8. According to the results of the two-factor interaction on trade-offs/synergies (Figure 9), the interaction between NL and LULC significantly impacted the relationship between bird plumage color CES and recreational value, followed by NL and DTW (Figure 9d). More importantly, the interaction between NL and other variables all exhibited substantial impacts on the trade-offs/synergies between bird plumage color CES and aesthetic value, with influences all exceeding 0.40 (Figure 9e).

## 4. Discussion

### 4.1. Park Landscape Characteristics Under the Trait-Based ES Framework

Our findings indicate that areas with a high richness of bird feeding guilds and significant positive contributions of effect traits were concentrated in regions with high vegetation cover, dense plant canopy, and low concentrations of impervious surface cover. In other words, parks provide superior supporting services, while birds contribute more effectively to cultural services in these areas. This aligns with previous studies [16,17]. Birds’ functional traits reflect their needs for park SES and the CES they provide to visitors. These traits are influenced by environmental elements such as vegetation and water bodies within parks [17,19]. The variations in park SES resulting from environmental differences are precisely mediated by avian response traits, while the expression of effect traits depends on the degree of adaptation of these response traits to their environment. For instance, the impervious open spaces in parks tend to attract visitors engaged in recreational activities, which causes disturbances and leads to a decline in the number of colorful birds, primarily insectivorous species [30]. Unlike shrubs and artificial structures, birds prefer arbors as shelter. In areas without arbors, birds are more likely to flee when disturbed [31]. To mitigate this issue, increasing the number of arbors around parks could attract a broader diversity of species with varying diet preferences, including visually appealing species. This strategy could effectively enhance both the SES provided by parks and the CES offered by birds. Additionally, the extent of visitor-bird interaction should also be taken into account [16].

### 4.2. ES Trade-Offs/Synergies in Human-Bird Relationships of Urban Park

With regard to the cultural services provided by parks, many visitors perceive that areas offering recreational value also enhance social interactions with others [14]. We speculate that these two services—recreational value and social relationships—have a synergistic effect.

Our results revealed that the spatial pattern of bird plumage color CES and aesthetic value mirrored that of park SES and aesthetic value, both dominated by trade-off relationships. This indicates that the CES (including those provided by the park and birds) has not been fully utilized in the park, and visitors cannot experience aesthetic values in most areas simultaneously. The CES generated by avian effect traits require supporting avian response traits. However, areas within parks that provide high levels of CES may not necessarily represent suitable avian habitats. This conclusion confirms that the cultural values of bird effect traits are closely linked to the avian ability to adapt to habitat conditions [16]. Therefore, enhancing the conditions that enable bird response traits to match environmental requirements should be prioritized. Another key finding was the positive correlations between park SES and recreational value, as well as park SES and social relationship value. Furthermore, the spatial range of synergies was larger than the trade-off for both ES pairs—park SES with recreational value and park SES with social relational value. In addition, the trade-offs/synergies spatial patterns of these ES pairs were similar. These findings may be attributed to the fact that most visitors come to the park with family or friends or join social activities in groups.

According to our study, noise level (NL) was the primary driver of the trade-offs and synergies between bird plumage color-related CES and aesthetic value, as well as between park SES and aesthetic value, consistent with our hypothesis. Interestingly, the noise we measured, primarily from visitors’ recreational activities, impacts both the distribution of birds and visitors’ aesthetic experiences. This disturbance may be exacerbated for visitors, as unlike birds, which must adapt to rare habitats [32], people freely and subjectively choose their viewing sites. The peak noise level in Xingqing Palace Park could reach 75.79 dB, which exceeds the 50 dB threshold for most urban birds [33]. Notably, we demonstrated that recreational value and park SES exhibited a synergistic relationship, despite the noise from recreational activities affecting other ES. This may be because only a subset of recreational activities generates noise, which in turn affects other ecosystem services. In summary, human intervention at the source to maintain noise levels within acceptable thresholds, or increasing tree density and height at the boundaries of noisy areas in future park management, could effectively filter noise levels [9]. Notably, much of the space in Xingqing Palace Park, where visitors and birds overlap, is located in areas with dense vegetation near the park’s main roads. These regions exhibited synergistic relationships between park SES and both recreational and social relationship values. As mentioned earlier, avian feeding guilds are particularly rich in areas with high vegetation cover. Additionally, we found that visitors also favor these areas. This is consistent with previous findings, as these areas with large trees provide food and safety for birds, shade for visitors, and make recreational activities more comfortable [34]. Thus, land use and land cover (LULC) and distance to roads (DTR), when interacting with noise levels (NL), influence the trade-offs/synergies between park SES and recreational value, as well as park SES and the social relationship value. Moreover, we found that avian adaptation to habitats also depends on the interaction between NL and LULC, distance to water (DTW), and DTR. Specifically, habitats with dense vegetation or proximity to water bodies supported diverse feeding guilds of colorful insectivorous birds, thus enhancing cultural ecosystem services. This coincides with previous studies [16,19]. Based on our observations, visitors appreciate classical architecture and flowering herbs in areas with high impervious cover or enjoy flowers in shrubby areas. Conversely, these areas are distant from the waterbody, lack trees, and experience higher summer temperatures. These conditions also increase predation risk for birds due to limited cover [35,36]. Therefore, we recommend planting shade-tolerant ornamental ground cover in forested areas to enhance the park’s aesthetic value in future park designs. To attract colorful insectivorous birds for roosting and nesting, we also suggest planting tall trees in areas with dense ground cover [37]. Reducing pesticide use and increasing the number of small water features, located away from the waterbody, will improve bird visibility and further enhance the park’s beauty.

### 4.3. Limitations

Compared to natural habitats, urban ecosystems support lower avian species diversity and exhibit increased niche similarity [38]. Consequently, the number of functional guilds is limited, while their species richness and abundance are totally different. Due to the constraints of avian functional groups at our study site, we selected diet traits exclusively as a proxy for SES. Nonetheless, diet traits—being one of the most fundamental categorizations of ecological niches—have been identified in previous studies as the only functional trait among various species characteristics that exhibits a significant negative correlation with human disturbance [39]. As a key functional trait, diet traits are crucial for understanding species’ habitat preferences in response to human disturbance, which is a focus of this study. It should be noted that the current analysis of plumage-based ecosystem services has two inherent limitations: (1) it does not consider sexually dimorphic traits, and (2) it overlooks seasonal variations in feather characteristics. Future studies should incorporate other functional traits such as nesting-site guilds and song diversity into the analysis to comprehensively evaluate the supporting services provided by parks for birds as well as avian CES for people. Despite these limitations, our findings offer valuable strategies for optimizing the management of urban parks and may be applied to other small-scale urban green spaces.

Due to limitations in human and material resources, our study only piloted the bird trait-service framework in a single park, identifying some landscape features that determine human–bird interactions in the park. However, the research employed only a limited set of bird traits (diet habits and plumage color), thereby focusing solely on the corresponding SES and CES related to these traits, which restricts the generalizability of the findings. Future research should expand both the number of study sites and the variety of bird traits to develop a more comprehensive human–bird synergy system based on an enhanced trait–service framework.

## 5. Conclusions

This study aimed to quantify the habitat quality (park SES) and aesthetic value provided by birds inhabiting a park (bird plumage color CES) through functional traits, and to explore the trade-offs and synergies, as well as the factors influencing these ES and multiple CES provided by the urban park at a grid scale. Our findings indicate that avian cultural ecosystem services and recreational value, as well as park supporting ecosystem services and recreational value, both exhibit synergistic relationships. Areas near the park boundaries and water bodies were found to exhibit synergistic attenuation, while areas with dense vegetation cover along main roads demonstrated synergistic promotion. In contrast, avian cultural ecosystem services and aesthetic value, as well as park supporting ecosystem services and aesthetic value, both displayed trade-off relationships. The areas experiencing trade-offs were the residents’ fitness plazas, which are subject to significant human disturbance, and the waterbody center. The primary negative factor influencing these four ES connections was the interaction between noise level and land use/land cover.

In terms of implications for urban parks with various visitors and bird populations, noise reduction and increased vegetation cover are essential, while ensuring that visitors can still enjoy their activities. Planting tall arbor trees in the park, particularly along its boundaries, expanding green separation zones around the perimeter, and incorporating small-scale water features can help reduce noise levels and create a more comfortable atmosphere. These improvements will enhance park supporting ecosystem services and attract more urban birds with diverse diet traits and visually appealing features, further boosting avian cultural ecosystem services.

Overall, our research provides a scientific reference for comparative case studies in urban parks, which can be applied to ES management. It is crucial to increase the diversity of colorful insectivorous birds in urban parks, as they offer a range of aesthetic and recreational opportunities, thus enhancing visitors’ well-being. Managers must clarify the trade-offs and synergies among ES, identify the critical factors influencing them, and conduct ecological planning in sub-areas in a rational and scientific manner. Areas experiencing synergistic attenuation and trade-offs should be prioritized for construction and management, while areas demonstrating synergistic promotion should be prioritized for conservation and protection.

## Figures and Tables

**Figure 1 animals-15-02619-f001:**
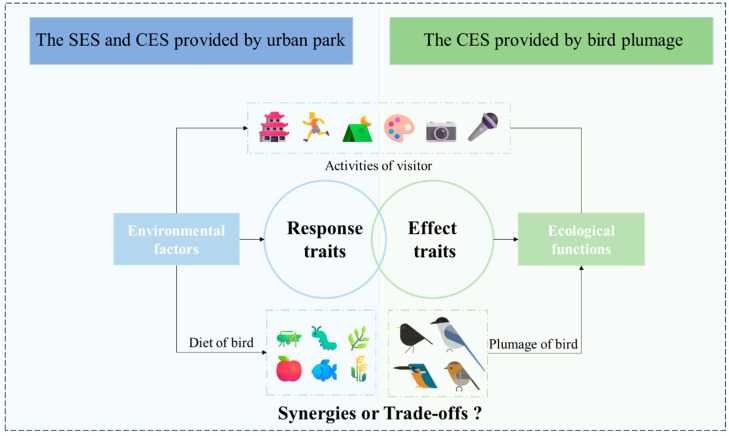
Trait-based ecosystem services framework [23], Source of drawing materials: https://www.jevaart.com (accessed on 17 July 2025).

**Figure 2 animals-15-02619-f002:**
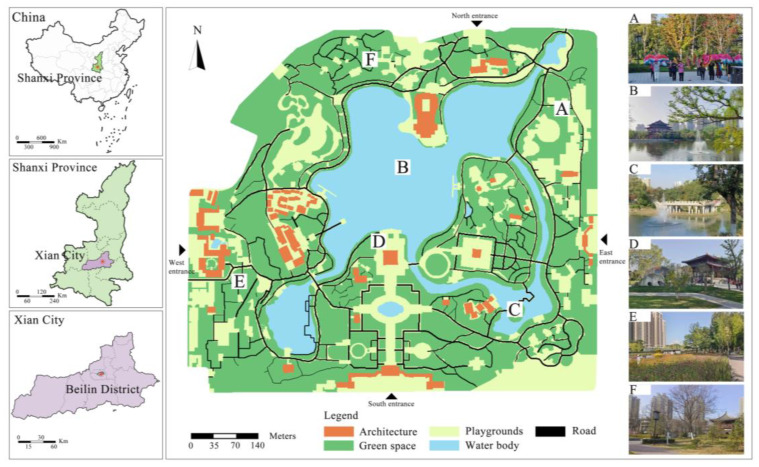
Study of the regional plan and main scenic spots.

**Figure 3 animals-15-02619-f003:**
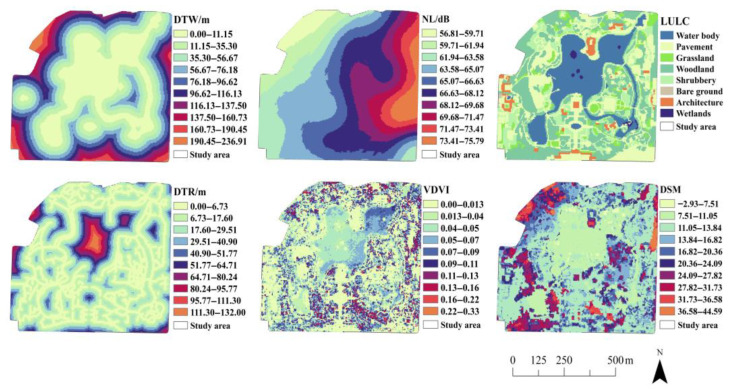
Environmental index data layers.

**Figure 4 animals-15-02619-f004:**
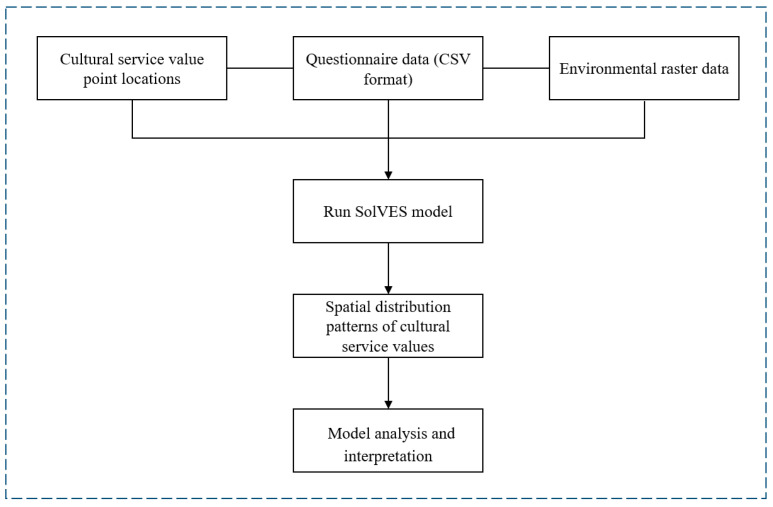
Technical workflow for cultural services assessment.

**Figure 5 animals-15-02619-f005:**
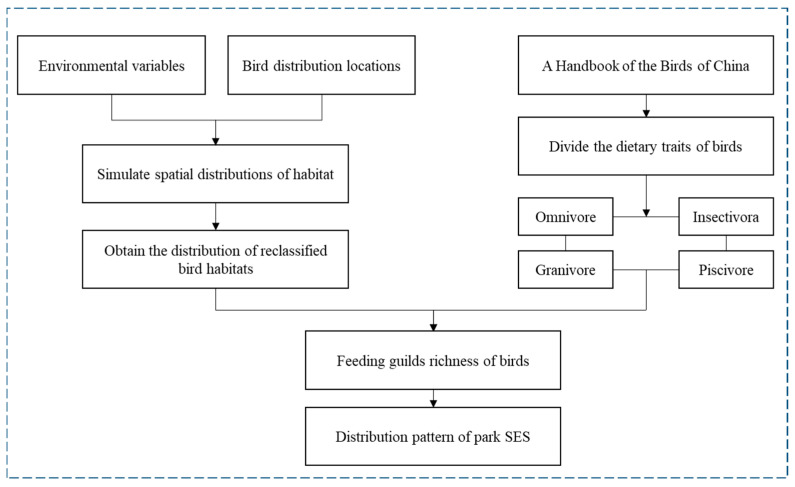
Park supporting ecosystem services calculation flowchart.

**Figure 6 animals-15-02619-f006:**
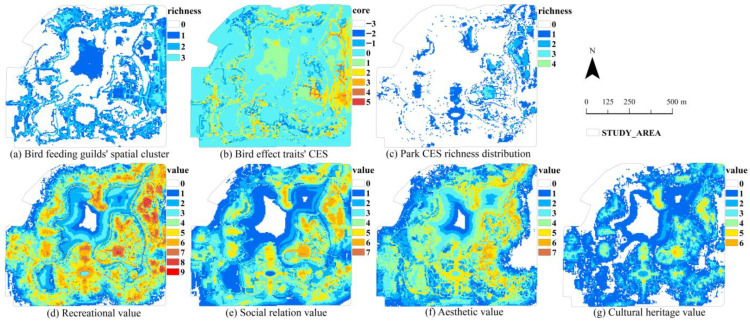
Spatial patterns of SES and CES.

**Figure 7 animals-15-02619-f007:**
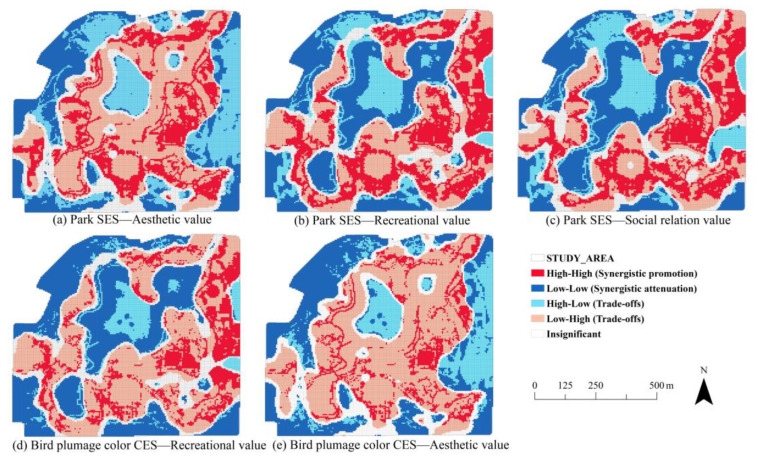
The trade-offs and synergies of ES at grid scale in Xingqing Palace park.

**Figure 8 animals-15-02619-f008:**
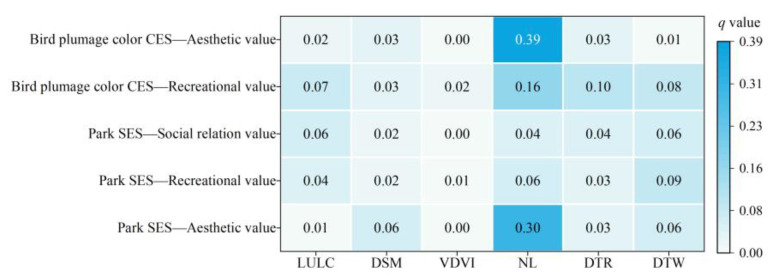
Effects of individual factors on trade-offs/synergies of different ES.

**Figure 9 animals-15-02619-f009:**
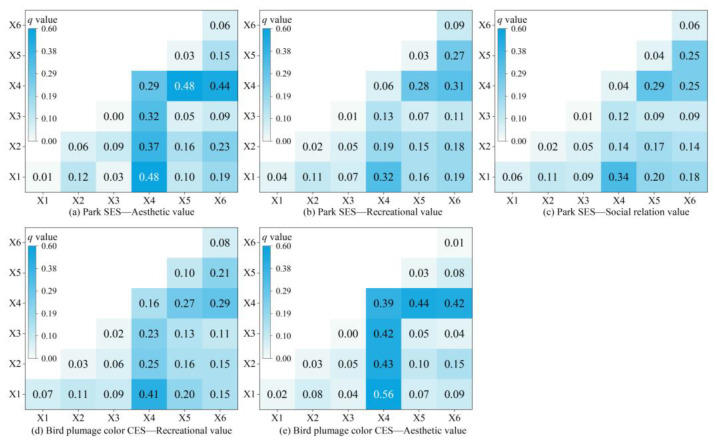
Effects of the two-factor interaction on trade-offs/synergies of different ES. X1: LULC, X2: DSM, X3: VDVI; X4: NL, X5: DTR, X6: DTW.

**Table 1 animals-15-02619-t001:** Definitions of CES * employed in the on-site survey.

Cultural Ecosystem Services	Definition
Social relation value	Here we offer dedicated social venues for gatherings with friends and family
Recreational value	Here we provide you with outdoor activity and recreational facilities
Aesthetic value	Here we provide scenic viewing areas for your enjoyment
Education value	Here we provide nature education facilities for ecological learning and science outreach activities
Cultural heritage value	Here are preserved historical artifacts that offer a glimpse into the past
Inspiration	Here we provide innovation-inspiring spaces designed to spark new ideas and creativity
Sense of place	Here we foster meaningful connections and cultivate a sense of belonging that lingers in your memory

* Defined by MEA, 2005 [2].

**Table 2 animals-15-02619-t002:** Data in the SolVES, MaxEnt, geographical detector analysis.

Data Name	Variable	Description
Boundary of the study area	-	The land cover classification was conducted through ArcGIS10.4 digitization based on drone imagery (with a spatial resolution of 1 m), following standardized land use classification criteria
Social survey points layer	-	Mapped by respondents and digitized through ArcGIS
Environmental variables	LULC	Using the vectorization tool in ArcGIS, UAV imagery was used as the basis for manual interpretation according to the land cover classification system
	DTW	Did Euclidean distances for roads, water bodies, and other types of wetlands using ArcGIS
	DTR	types of wetlands using ArcGIS
	VDVI	Processing of UAV remote sensing imagery containing visible light bands employing ENVI 5.6 and calculations using the Green Leaf Index = (b1 ge 0.04) × b1 + (b1 lt 0.04) × 0
	* DSM	Includes the height of surface vegetation and houses. Each UAV imagery contains one DSM value, stitched together using the software Pix4D to form a DSM of the entire UAV imagery. Then, ArcGIS is used to crop out the excess to obtain a DSM of the entire study area with a resolution of 5 × 5 m
	NL	Noise maxima were measured at survey sample points on the bird survey sample line using a Sound Level Meter, and then processed through the multi-ring buffer, Kriging Interpolation in ArcGIS

* The DSM among the environmental variables was only used in the MaxEnt model and Geographical Detector analysis, while all other environmental variables were included in the SolVES model, MaxEnt model, and Geographical Detector analysis.

**Table 3 animals-15-02619-t003:** M-VI and mean nearest-neighbor analysis.

CES Value Type	* N_COUNT	R_RATIO	Z_SCORE	M-VI
Social relation value	89	0.63	−6.75	7
Recreational value	123	0.44	−11.79	9
Aesthetic value	100	0.47	−10.12	7
Education value	56	0.48	−7.40	4
Cultural heritage value	63	0.49	−7.70	6
Inspiration	20	0.52	−4.11	2
Sense of place	19	0.57	−3.57	2

* The N_COUNT represents the total number of social value points marked by respondents; R_RATIO is the average nearest neighbor ratio; Z_SCORE denotes the standard deviation; and M-VI stands for the Maximum Social Value Index.

## Data Availability

The original contributions presented in this study are included in the article/Appendix A. Further inquiries can be directed to the corresponding author.

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
