# Peer review of "Park Visitors and Birds Connected by Trade-Offs and Synergies of Ecosystem Services"

_animals, 2025, doi:10.3390/ani15172619_

Round 1

Reviewer 1 Report

Comments and Suggestions for Authors

The paper is a good case study (it is a novelty) to use birds and other environmental variables for evaluation of urban parks for rationale management. The statistical analysis is robust and maps are  generated with high precision. In overall, very good presentation of results.

As an ornithologist, I focus my attention on the birds. I have found few points which can be improved. First of all, the authors should clearly explain why they have choosen only some traits out of may others, e.g. only feeding guilds were taken into analysis, why not nesting-sites guilds (tree/shrubs, ground, tree holes, buildings), residency (migratory, resident, irregular visitors, etc.); only plumage colour, why not bird size, their songs, abundance). Those other options should be mentioned in the methodology, and on this background, your selection should be outlined (perhaps with some rationale for you choice). 

The line transect method has been used to count birds. This method is sufficient for this type of study, but more details should be provided here. How many transects were designed, what was their average and total lengthDi. Were birds counted once only on each transect, or 2-3 times on each trasect?, etc.  

Table S2, what is ment by 'individual count'. I presume: number of individuals recorded. In the table the total should be given and dominance (percentage of the number of individuals of a given species related to the total number of all individuals of all species) calculated for each species. Aslo each species should be here 'affiliated' to the specific feeding guild (in a separate column). Hirundo rustica Latin name is Barn Swallow in English, not just 'Swallow'   

Reviewer 2 Report

Comments and Suggestions for Authors

I found your study to be very interesting and is on a topic that I think will be of interest to many. I applaud you for going beyond simple correlations between human activity and bird diversity, and instead trying to tease apart the mechanisms and consequences of park use by humans and park use by birds. Your methods are sound, and your analyses seem appropriate and well described. Your conclusions follow logically from your results. Your paper is well written, and I only had a handful of minor stylistic edits. The only place where I thought you needed to expand upon was in the introduction; you need to define cultural, regulating, and support ecosystem services. I wish you well in your efforts to publish your work.
